

# Modeling and implementation of a real-time digital twin for the Stewart platform with real-time trajectory computation

Xiurui Ding[1] and Jinsheng Xing[2]

[1] Aerospace Manufacturing, Faculty of Engineering and Applied Sciences, Cranfield University, UK, United Kingdom
[2] School of Mathematics and Computer Science, Shanxi Normal University, Taiyuan, Shanxi, China

## ABSTRACT

The concept of a digital twin is increasingly acknowledged as an innovative and promising tool with significant potential in various end-use applications. At the heart of digital twin technology is the acquisition of real-time data from physical entities. However, the occurrence of disturbances necessitates the incorporation of resilience features within the digital twin architecture. The primary objective of this article is to develop resilient digital twins specifically for the Stewart platform. This work focuses on constructing the virtual component of the digital twin using MATLAB/Simulink and subsequently integrating this virtual model with its physical counterpart to establish a comprehensive digital twin system. Unlike other models, this system includes a motion trajectory computation module. This module is designed to receive signals from physical entities and convert them into motion trajectory data for input into the model, thereby aiming to accurately reflect the state of the physical entities under disruptive conditions. This functionality significantly enhances the reliability of the system beyond that of traditional digital twin systems. Furthermore, the article explores novel strategies and a framework for enhancing the resilience of the Stewart platform to disturbances.

# INTRODUCTION

## Research background

The digital twin serves as a virtual replica of a physical entity, continuously synchronizing real-time data with its physical counterpart to mirror its state accurately (*Singh et al., 2021*). As a tool capable of reflecting the real-time status of its target, the credibility of a digital twin constitutes a fundamental determinant of its overall effectiveness (*Song et al., 2023*). One essential criterion for assessing the trustworthiness of a digital twin is the capability of the systems to detect disturbances or interruptions and to make resilient decisions in response (*Eirinakis et al., 2022*). Digital twins that fulfill this criterion prove more adept at preemptively monitoring and optimizing the behavior of the physical entity they represent. This resilience allows digital twins to offer a more robust and reliable

Corresponding author
Jinsheng Xing,
xjs19640408@163.com

reflection of their physical counterparts, even under adverse conditions. In industries such as manufacturing and precision engineering, where operational reliability and efficiency are paramount, this level of resilience becomes essential. The ability of digital twins to anticipate and respond to potential disruptions ensures they can continuously provide accurate insights and drive optimized decision-making. Consequently, the development of resilient digital twins is poised to become a focal point within the field of digital twin research. The recognition of the resilient digital twin to disturbances or interruptions provides an exact representation of the state of the physical entity in the presence of disturbances, and at the same time makes the entire digital twin system more trustworthy. This focus is particularly pertinent for applications in industrial manufacturing and precision equipment, where the ability to anticipate and mitigate potential disruptions is paramount (*Vrabič et al., 2021*).

## Digital twin

Digital twins have found applications across a diverse array of fields, encompassing simulations of spacecraft and wind turbines, as well as modelling of chemical processes and urban planning. The concept of digital twins was introduced by Grieves, having in its basic form three parts: physical products in real space, virtual products in virtual space, and the connection of data and information that ties the virtual and real products together (*Eirinakis et al., 2022*). This technology serves as a predictive method that relies on numerical models, which are perpetually refined within a virtual environment. Distinct from traditional numerical models, digital twins facilitate the evaluation of all conceivable decisions within the virtual realm, thereby exerting a more pronounced impact on real-world outcomes (*Van Der Aalst, Hinz & Weinhardt, 2021*). The capacity to manipulate real-world outcomes *via* virtual models positions the digital twin as a pivotal facilitator of Industry 4.0 within the manufacturing sector, particularly in the realm of manufacturing strategy optimization (*Eirinakis et al., 2022*). For instance, a specific study presents a reconfiguration framework that integrates digital twins and artificial intelligence into manufacturing systems. The framework was used to simulate a manufacturing process with multiple industrial robots performing various tasks to determine the optimal configuration of the manufacturing system. However, this digital twin framework is optimized based on data from the physical side and cannot recognize sudden changes in the physical side. Digital twin models may no longer be trustworthy when anomalies occur on the physical side. Therefore, cognitive digital twins (CDTs) are an evolutionary approach that utilizes services and tools to implement human-like cognitive capabilities in DTs to enable manufacturing systems to recognize and handle abnormal and disruptive events in the production process and execute decisions to mitigate their consequences (*Eirinakis et al., 2022*). This ability to enable a system to maintain or quickly return to a stable state during and after a major accident, or under sustained significant stress, was defined as resilience (*Leng et al., 2023*).

The concept of the resilient digital twin is designed to minimize losses during disruptions and facilitate a swift recovery to steady-state operations post-disruption. The

development of resilient digital twins emphasizes early disruption detection, impact assessment, root cause analysis, and the support of decision-making processes to mitigate the consequences of such disruptions (*Tang, Emmanouilidis & Salonitis, 2020*). Some previous studies have contributed to this field by providing a comprehensive framework for the design and implementation of digital twins. Their research explores the potential for integrating diverse models under the umbrella of digital twins within the manufacturing context. Utilizing the specific example of a weld melt pool, their framework incorporates both forward and inverse prediction models within a real-time digital twin module. This integration is aimed at predicting nodal and peak temperatures, thereby enhancing the resilience and flexibility of the manufacturing process (*Papacharalampopoulos, Michail & Stavropoulos, 2021*). Building upon this foundational framework, *Eunike et al. (2022)* developed a decentralized scheduling system designed to be resilient to disruptions on the shop floor. The study initiates with an introduction to the concept of digital twins for real-time flow shop assembly systems. It then progresses to an examination of how the digital twin model can facilitate the creation of communication protocols within the system. Additionally, the study explores the development of local scheduling optimization strategies. These strategies are aimed at enhancing the resilience of the system to disruptions, thereby ensuring more robust operational continuity (*Eunike et al., 2022*). As research advances, the application of digital twins and their resilience attributes is progressively being extended to additional domains. For instance, recent studies have begun to integrate resilient digital twins within the civil infrastructure sector, aiming to aid infrastructure managers in enhancing the sustainability and resilience of their assets. This article develops and validates a digital twin model of a railway network, affirming its resilience. Furthermore, it establishes a framework that will guide future research on recoverable digital twin models applicable to railway and road networks (*Vieira et al., 2022*). Concurrently, the utilization of digital twins in healthcare has emerged as a prominent research avenue for applications of resilient digital twins. *Zhang et al. (2021)* recognized cyber resilience as a crucial element for enabling precision medicine through healthcare digital twins, with vulnerability detection identified as a fundamental technique for enhancing cyber resilience. Their study introduces a novel end-to-end cyber resilience scheme designed to identify potentially vulnerable features within healthcare digital twin software projects. The efficacy of this scheme is validated through its application in a lung cancer diagnosis example, demonstrating its potential utility in improving the security and reliability of digital twin technologies in medical contexts.

Research concerning recoverable digital twin models in the aforementioned areas has primarily been confined to the development of digital twin frameworks and the verification of their recoverability. In stark contrast, applications within the industrial manufacturing sector have experienced substantial advancements (*Khan, 2022*). In this domain, recoverable digital twins have demonstrated their capability to exert influence over and control the physical components of the model, marking significant progress in the integration and functionality of digital twins in manufacturing processes. A recent study has introduced a methodological approach for modelling and controlling the elastic

dynamics of a reconfigurable electronic assembly line when faced with disruptions. This method utilizes a digital twin model of the electronic assembly line2 and integrates an open reconfigurable architecture (ORA) to facilitate the assembly reconfiguration of lines. It identifies and utilizes the time delays associated with interruptions to characterize their spatio-temporal properties. Furthermore, the study employs a systematic approach based on maximal additive algebra for modeling the resilient dynamics amid disruptions. A resilience control strategy for digital twin platforms aimed at minimizing production losses has been developed and empirically tested on a smartphone assembly line. The effectiveness of this strategy is confirmed through a comparative analysis, demonstrating its viability in enhancing production resilience (*Zhang et al., 2021*). Additionally, there are investigations that have introduced a novel methodology for assessing and enhancing the robustness and resilience of digital twins within manufacturing processes. These studies have developed an intelligent agent-based architecture that can detect disturbances to the digital twin system, evaluating these disturbances, determining appropriate countermeasures, and adapting the digital twin system accordingly. A simulation-based case study illustrates that the robustness and resilience of the digital twin model can be significantly improved in scenarios characterized by uncertainties and disturbances (*Vrabič et al., 2021*). However, the majority of recoverable digital twins in industrial manufacturing predominantly concentrate on simulating manufacturing processes and assembly lines, namely the construction of digital twin models that encompass the entire manufacturing system. There exists a limited number of studies that focus on modeling individual components or modules within the manufacturing system.

## Stewart platform

The Stewart platform, a six-degree-of-freedom parallel robot, epitomizes a complex mechanical system designed to perform advanced motion simulations, precision manipulations, and active vibration control. Constructed using six linear actuators—pneumatic, hydraulic, or electric—the platform is intricately connected to both top and bottom plates through ball or gimbal joints. As shown in Fig. 1.

This configuration is pivotal in facilitating its capacity to deliver six degrees of freedom (DoF), encompassing three translational motions (along the x, y, and z axes) and three rotational movements (pitch, roll, and yaw). The extensive range of motion of the Stewart platform is accomplished through either independent or coordinated actuation of its six legs. This arrangement enables the execution of complex and precise movements, surpassing the capabilities typically achievable with more conventional mechanical systems. The ability to control each leg independently allows for sophisticated manipulation and adjustment of the platform's posture and position, making it an invaluable tool in fields requiring high precision and versatility (*Dasgupta & Mruthyunjaya, 2000*).

The Stewart platform, characterized by its high degree of freedom, flexibility, and precision, is versatile enough for a myriad of applications. Originally engineered to simulate aircraft flight attitudes, the platform has seen extensive innovation and enhancement in its control strategies to enhance its simulation capabilities. In a particular

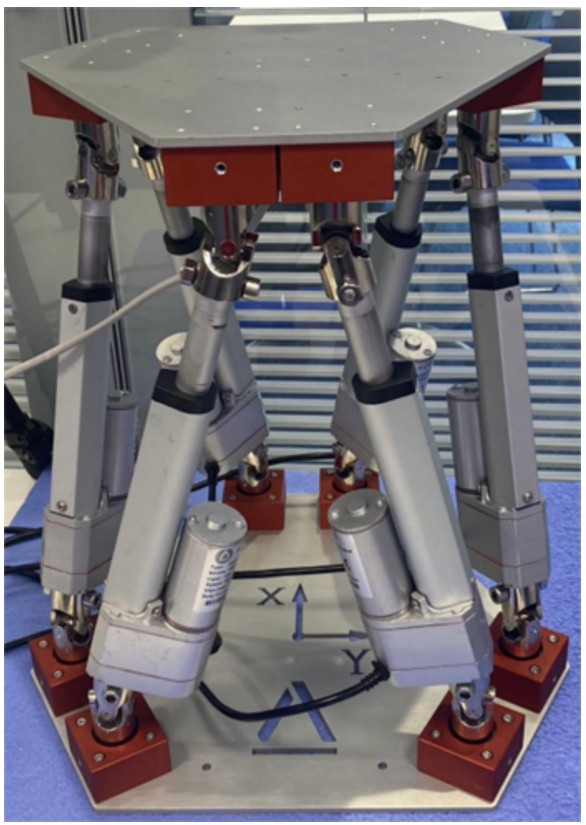

**Figure 1  Stewart platform.**

study, a flight simulator incorporating the Stewart mechanism was equipped with a feedback linearization scheme and a state-dependent proportional derivative controller. This configuration was designed to provide more realistic visual and vestibular flight feedback, aiming to closely replicate actual flight conditions (*Eftekhari & Karimpour, 2018*).

Furthermore, another study focused on augmenting the accuracy of the Stewart platform in simulating flight attitudes by integrating a sliding mode control strategy (SMS). This control approach was introduced to improve the robustness of the platform against modeling uncertainties and external disturbances, thereby enhancing its performance in delivering precise flight attitude simulations (*Velasco et al., 2020*). These modifications underscore the ongoing efforts to refine and expand the applications of the Stewart platform within simulation environments, leveraging its dynamic control capabilities to achieve higher fidelity in simulations. Owing to the inherent advantages of the Stewart platform in attitude simulation, its applications extend beyond aircraft attitude simulation. A specific study has introduced a large-scale trimming platform, leveraging the design principles of the Stewart platform to meet the trajectory tracking requirements of large spherical radio telescopes. This adaptation involves dynamic, and singularity analyses conducted *via* the Jacobian matrix. These analyses are critical in validating the effectiveness of the finely tuned Stewart platform in controlling the trajectory tracking of feeders for large spherical radio telescopes (*Su & Duan, 2000*).

The study demonstrates how modifications to the traditional Stewart platform can enhance its utility in precision tracking applications, emphasizing its capability to handle complex, large-scale engineering challenges. Furthermore, researchers are exploring the application of this high-precision instrument in more compact scenarios. For instance, several studies have employed the Stewart platform for precision microsurgery across various medical disciplines. One study highlighted the capability of the six-degree-of-freedom system to accurately position an endoscope during neuroendoscopic procedures. This demonstrates the platform's potential in enhancing surgical precision and safety, thereby significantly contributing to the field of minimally invasive surgery (*Wapler et al., 2003*). The ability to precisely control the position and orientation of surgical instruments like endoscopes is pivotal in complex medical interventions, underscoring the versatility and utility of the Stewart platform in medical applications.

As previously noted, the high precision of the Stewart platform underscores its significance in the field of engineering. Consequently, there is an increasing focus among researchers on controlling and monitoring the operations of this high-precision platform (*Bingul & Karahan, 2012*). In line with these developments, the construction of a digital twin model of the Stewart platform has emerged as a prominent research trend. This approach facilitates enhanced analysis, simulation, and optimization of the performance of the platform, thereby advancing its application in complex engineering scenarios. The digital twin model not only replicates the physical attributes and dynamics of the Stewart platform but also enables predictive maintenance and real-time monitoring, which are crucial for extending the platform's operational life and enhancing its efficiency (*Walica & Noskievič, 2024*). Although research on digital twins for the Stewart platform predominantly concentrates on its applications, recent studies have begun exploring control methods for these digital models. For instance, *Camacho, Medrano & Carvajal (2020)* utilized digital twins to validate the localization processes in a vehicle driving simulator based on the Stewart platform, showcasing the practical applications of these virtual models in enhancing system verifications.

Moreover, in a significant shift towards advanced control strategies, *Walica & Noskievič (2024)* incorporated the hardware-in-the-loop (HiL) technique into the control design of digital twin models for the Stewart platform. This study commenced with validating the control algorithm through model-in-the-loop (MiL) simulations. Subsequently, the validated code was deployed to a PLCnext controller within a MATLAB/Simulink environment, facilitating a seamless transition from design to integration and testing phases (*Walica & Noskievič, 2022*).

### Research aims

In conclusion, the primary goal of this article is to develop a digital twin system for the Stewart platform and to propose a resilient framework based on the digital twin system. In this article, a virtual component is constructed within Simulink to facilitate a

resilient digital twin system tailored for the Stewart platform. This component efficiently recognizes signals and data transmitted by the physical entity and processes these signals into motion data using a specifically designed trajectory computation module. The system is thus enabled to accurately reflect the dynamic state of physical entity.

## METHODOLOGY

This project comprises two primary components: the construction of a virtual model and the integration with a physical Stewart platform. A multibody dynamics model is first developed in Simulink, including modules for inverse kinematics, Jacobian computation, and trajectory generation. This virtual model is then connected to a physical StewartPro robot from Acrome Robotics *via* customized communication protocols, enabling real-time data exchange as shown in Fig. 2. The system allows the simulation to track the motion of the physical platform in real time, thereby completing the virtual segment of the digital twin. The trajectory module computes motion commands based on input parameters or real-world data and feeds them into the virtual model. The digital twin framework, developed in MATLAB/Simulink, is aligned with the physical platform to realize a functional digital twin.

### Kinematic analysis and modelling
#### Kinematic inverse solution
The kinematic inverse solution is a method used to figure out what movements (like joint angles) a robot or mechanical system needs to make to reach a specific position or orientation in space. The kinematic inverse solution for the Stewart platform involves a computational process designed to ascertain the required lengths of each support arm to achieve a specified platform attitude and position (*Bingul & Karahan, 2012*). Initially, coordinate systems for both the static platform (lower platform) and the moving platform (upper platform) are established at the platform's initial position and attitude. Subsequently, spherical coordinate systems corresponding to both the static and moving platforms are derived. This foundational step facilitates the subsequent calculations necessary to control the platform's spatial orientation and positioning accurately.

Assuming the spherical coordinate systems of the static and moving platforms are respectively: Static platform (lower platform) coordinate system:

$$\vec{B_1}, \vec{B_2}, \vec{B_3}, \vec{B_4}, \vec{B_5}, \vec{B_6}.$$

Moving platform (upper platform) coordinate system:

$$\vec{P_1}, \vec{P_2}, \vec{P_3}, \vec{P_4}, \vec{P_5}, \vec{P_6}.$$

Subsequently, it is essential to configure the displacement and rotation matrices for the Stewart platform. Assuming that the expected position coordinates of the moving platform

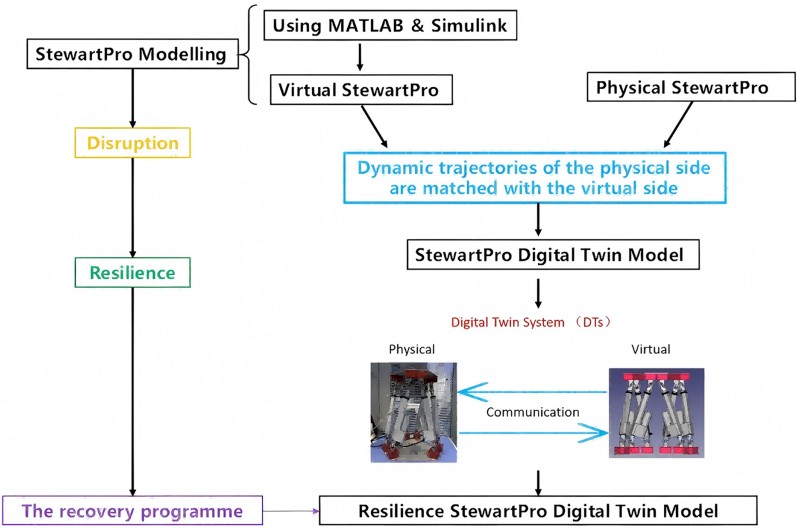

**Figure 2 Research approach.** The different colors in the figure represent distinct components of the digital twin system: the physical Stewart platform (blue), the virtual twin developed in MATLAB/Simulink (green), and the bi-directional communication between them (red). These elements collectively form the resilient digital twin framework.

are (x, y, z), and the Euler angle of the expected attitude (θ, φ, ψ) (pitch, roll, yaw). Let the displacement matrix be:

$$\vec{P} = \begin{bmatrix} x \\ y \\ z \end{bmatrix}.$$

And the rotation matrix is:

$$R = R_z * R_y * R_x.$$

The rotation matrix is calculated as follows:

$$R_x = \begin{bmatrix} 1 & 0 & 0 \\ 0 & \cos(\theta) & -\sin(\theta) \\ 0 & \sin(\theta) & \cos(\theta) \end{bmatrix}$$

$$R_y = \begin{bmatrix} \cos(\varphi) & 0 & \sin(\varphi) \\ 0 & 1 & 0 \\ -\sin(\varphi) & 0 & \cos(\varphi) \end{bmatrix}$$

$$R_z = \begin{bmatrix} \cos(\psi) & -\sin(\psi) & 0 \\ \sin(\psi) & \cos(\psi) & 0 \\ 0 & 0 & 1 \end{bmatrix}.$$

Then the coordinates of the ball joint of the moving platform with respect to the static platform are:

$$\overrightarrow{P_{B_i}} = \vec{P} + R * \overrightarrow{P_i} \quad (i = 1, 2, 3, 4, 5, 6).$$

Then the bar vectors along the legs of the Stewart platform are:

$$\vec{l_i} = \vec{P_{B_i}} - \vec{B_i} \quad (i = 1, 2, 3, 4, 5, 6).$$

The above formula is used to calculate the length of each arm of the platform control to achieve the desired platform position and attitude.

### Jacobian matrix

The Jacobian matrix is a mathematical tool used in robotics, kinematics, and control theory to describe the relationship between the velocities (or rates of change) of a system's joints and the resulting velocity of its end-effector (the tool or hand of a robot, for example). It is a way to capture how changes in joint positions or angles affect the movement of the end-effector. The Jacobian matrix relates the velocities of the active joints (actuators) to the generalized velocity of the moving platform. Specifically, the Jacobian matrix is used to describe how variations in the length of each of the platform support arms affect the position and attitude of the platform (*Bingul & Karahan, 2012*). It converts the rate of change of arm lengths into linear and angular velocities of the platform.

The point of the Stewart platform Jacobian matrix is how to relate the coordinates (x, y, z) and attitude ($\theta$, $\varphi$, $\psi$) of the moving platform with respect to the base platform using the lengths obtained by inverse kinematics solution. For the Stewart parallel robot model, let $l_i$ be the length of the actuator i, and be the position vector from the origin O of the moving platform's centre-of-mass coordinate system to point $\vec{P}$, $\vec{\omega_o}$ and $\vec{v_o}$ are the angular and linear velocity vectors of the moving platform in the base coordinate system, respectively, and then the velocity vector at any point $\vec{P}$ on the moving platform is:

$$\vec{v_{P_i}} = \vec{\omega_0} \times \vec{r_{oP_i}} + \vec{v_0}.$$

Assume that $u_i$ is the unit vector of the $i$ joint and has a positional representation as:

$$\vec{P_{B_i}} = \vec{P} + R * \vec{b_i} = \vec{l_i} + \vec{a_i}.$$

Derivation on both sides of the equation yields:

$$\vec{v_p} + R * \vec{b_i} = l_i \vec{u_i} + l_i \vec{u_i}.$$

Due to the following equation:

$$R * \vec{b_i} = \vec{\omega_p} \times \left( R * \vec{b_i} \right) = \vec{\omega_p} \times \vec{b_{i,o}}$$

$$l_i \vec{u_i} = l_i \left( \vec{\omega_{i,o}} \times \vec{u_i} \right).$$

Then, the above equation can be transformed into

$$\vec{v_p} + \vec{\omega_p} \times \vec{b_{i,o}} = \vec{l_i} + l_i \left( \vec{\omega_{i,o}} \times \vec{u_i} \right).$$

It is known that $\vec{v_p} = \begin{bmatrix} \dot{x}_p & \dot{y}_p & \dot{z}_p \end{bmatrix}^T$ in order to eliminate $\vec{\omega_o}$, it is necessary to left-multiply each side of the equation by one $u_i$, thus obtaining:

$$\vec{u_i} \cdot \vec{v_p} + \left( \vec{b_{i,o}} \times \vec{u_i} \right) \vec{\omega_p} = l_i.$$

Additionally, for each of the six legs of the Stewart platform, the Jacobian matrix is derived as follows:

$$
J = \begin{bmatrix}
\vec{u}_1^T & \left(\vec{b}_{1,0} \times \vec{u}_1\right) \\
\vec{u}_2^T & \left(\vec{b}_{2,0} \times \vec{u}_2\right) \\
\vec{u}_3^T & \left(\vec{b}_{3,0} \times \vec{u}_3\right) \\
\vec{u}_4^T & \left(\vec{b}_{4,0} \times \vec{u}_4\right) \\
\vec{u}_5^T & \left(\vec{b}_{5,0} \times \vec{u}_5\right) \\
\vec{u}_6^T & \left(\vec{b}_{6,0} \times \vec{u}_6\right)
\end{bmatrix}.
$$

The matrix can be simplified as

$$
\dot{L} = J \cdot \dot{X}.
$$

From the derived Jacobian matrix, it is evident that the matrix effectively maps joint velocities (or drive velocities) to actuator velocities and orientations. In the context of the Stewart platform, the Jacobian matrix elucidates how the velocities of the six actuator arms influence the changes in position and attitude of the platform. This matrix plays a pivotal role in the subsequent dynamics calculations and the design of control algorithms, providing critical insights into the effects of force and torque. Specifically, how the force exerted by the actuators is converted into the requisite force or torque at the joints.

## Trajectory analysis and modelling

This section will demonstrate the process of analyzing the motion trajectories of the Stewart platform and constructing the aforementioned trajectory computation module in Simulink. By deriving the kinematic inverse solution for the Stewart platform, it becomes apparent that the motion trajectory is governed by the attitudes of the upper platform and the lengths of the six actuator arms (*Virgil Petrescu et al., 2018*).

The control parameters for the arm lengths of the Stewart platform are defined by the travel distances dx, dy, and dz along the x, y, and z axes, respectively. The control parameters for the attitude of the upper platform are specified by the Euler angles: pitch ($\theta$), roll ($\varphi$), and yaw ($\psi$). Consequently, the trajectory of the Stewart platform can be manipulated through these six parameters. The trajectory computation module, constructed based on this principle, is capable of calculating the rotation matrix and the arm lengths from these six parameters, thereby determining the trajectory. Upon integration with the physical entity, this module retrieves these six parameters from the data provided by the physical entity to reconstruct its trajectory. Subsequently, this trajectory data is fed into the inverse solution module and controller of the virtual model, enabling the simulation of the virtual model's movements.

The process of calculating the length of the arm using the parameters dx, dy, and dz is relatively simple and only requires subtracting the initial position coordinates from the final position coordinates of the arm with the following equation:

$$
L = \sqrt{dx^2 + dy^2 + dz^2}.
$$

Based on the kinematic inverse solution derived above, the length of the arm and the upper platform position for different trajectories of the Stewart platform can be calculated by the Simulink module.

The computation of the rotation matrix necessitates the utilization of three Euler angles as parameters. The rotation matrices corresponding to the roll, pitch, and yaw angles are specified in the derivation of the kinematic inverse solution, outlined as follows:

$$R_x = \begin{bmatrix} 1 & 0 & 0 \\ 0 & \cos(\theta) & -\sin(\theta) \\ 0 & \sin(\theta) & \cos(\theta) \end{bmatrix}$$

$$R_y = \begin{bmatrix} \cos(\varphi) & 0 & \sin(\varphi) \\ 0 & 1 & 0 \\ -\sin(\varphi) & 0 & \cos(\varphi) \end{bmatrix}$$

$$R_z = \begin{bmatrix} \cos(\psi) & -\sin(\psi) & 0 \\ \sin(\psi) & \cos(\psi) & 0 \\ 0 & 0 & 1 \end{bmatrix}.$$

However, the matrices represent the individual rotation matrices of the Stewart platform corresponding to each Euler angle. To compute the generating trajectory, it is necessary to calculate the composite rotation matrix derived from the combination of these three matrices. The composite rotation matrix is calculated by the following equation:

$$R = R_z * R_y * R_x$$

The composite rotation matrix is

$$R = \begin{bmatrix} \cos(\varphi)\cos(\psi) & -\sin(\psi)\cos(\varphi) & \sin(\varphi) \\ \sin(\varphi)\sin(\theta)\cos(\psi) + \sin(\psi)\cos(\theta) & -\sin(\varphi)\sin(\theta)\sin(\psi) + \cos(\theta)\cos(\psi) & -\sin(\theta)\cos(\varphi) \\ -\sin(\varphi)\cos(\psi)\cos(\theta) + \sin(\psi)\sin(\theta) & \sin(\varphi)\sin(\psi)\cos(\theta) + \cos(\psi)\sin(\theta) & \cos(\theta)\cos(\varphi) \end{bmatrix}.$$

Based on the above calculated composite rotation matrix, its computational module can be built in the MATLAB/Simulink environment. The module of this composite rotation matrix in Simulink.

In the architecture developed, the composite rotation matrix is encapsulated into discrete sub-modules, which are then integrated with the module responsible for computing the arm length. The integration culminates in the formation of a trajectory calculation module.

## Dynamical models of stewart platform

The three-dimensional dynamics model of the Stewart platform constitutes a critical component of the virtual representation in its digital twin architecture. In this project, the model was constructed using SolidWorks, a CAD software developed by Dassault, which is renowned for its robust capabilities in 3D modeling. A pivotal factor in selecting SolidWorks was its compatibility with MATLAB through the installation of the Simscape Multibody Link plug-in. This plug-in facilitates the transformation of 3D models from SolidWorks into dynamic models within Simulink, enabling co-simulation. The integration of these tools not only substantially reduces the workload but also enhances the

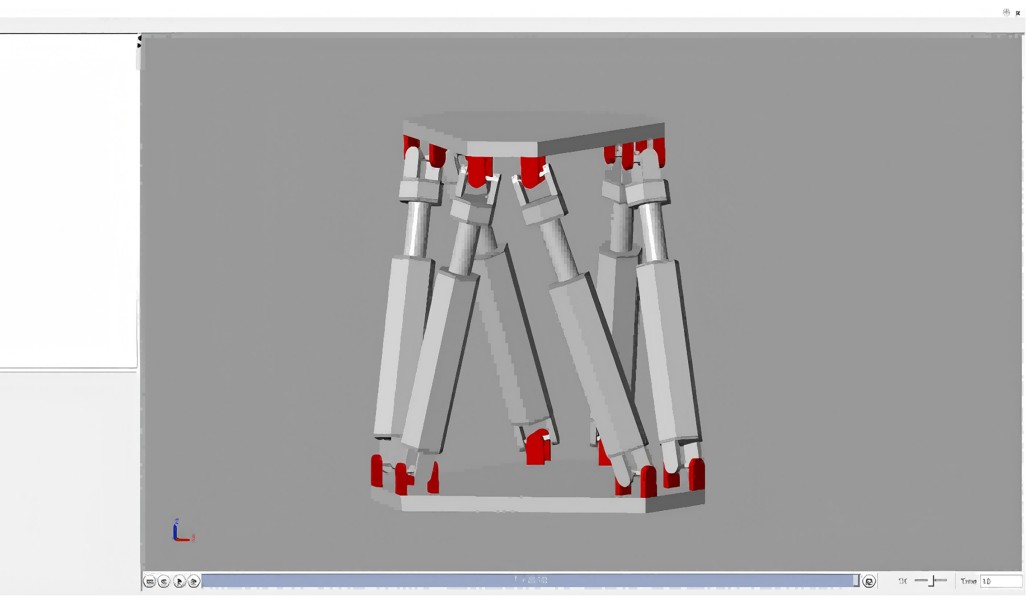

**Figure 3** **Stewart platform 3D model.**

accuracy of the models. Given that the 3D model provided by Acrome, which represents the physical entities of the Stewart platform, does not constitute an active assembly, it is incapable of being converted into a Simulink model suitable for dynamics simulation through the plug-in. Consequently, a custom-built Stewart platform was employed to facilitate the dynamics simulation and to construct the virtual counterpart. This model of the Stewart platform closely mimics the physical structure in form and possesses the capability to execute all typical motions associated with Stewart platforms.

By installing the Simscape Multibody Link plug-in in SOLIDWORKS, the 3D model of the Stewart platform is converted into a dynamics model in Simulink that can be used for simulation. Running the model allows the visualisation of Simulink to see the Stewart platform dynamical model as shown in Fig. 3.

## RESULTS AND DISCUSSION

### Model performance validation

As outlined in the research project plan, the simulation model of the Stewart platform will be employed for the virtual entities of the digital twin. Prior to integrating the physical entities, it is essential to validate the dynamics of the constructed multibody model. The primary aim of this validation is to fine-tune and ascertain the three proportional-integral-derivative (PID) controller gains: proportional, integral, and differential. Furthermore, it is imperative to verify that there is no excessive position error or oscillation when the model receives signals from the physical entity.

To validate the multibody dynamics model, it is imperative to compare the relevant physical states with their simulated equivalents. As no physical entities have been incorporated at this juncture, a predefined motion trajectory will be employed for validation purposes. The subsequent analysis will involve a detailed quantification of the

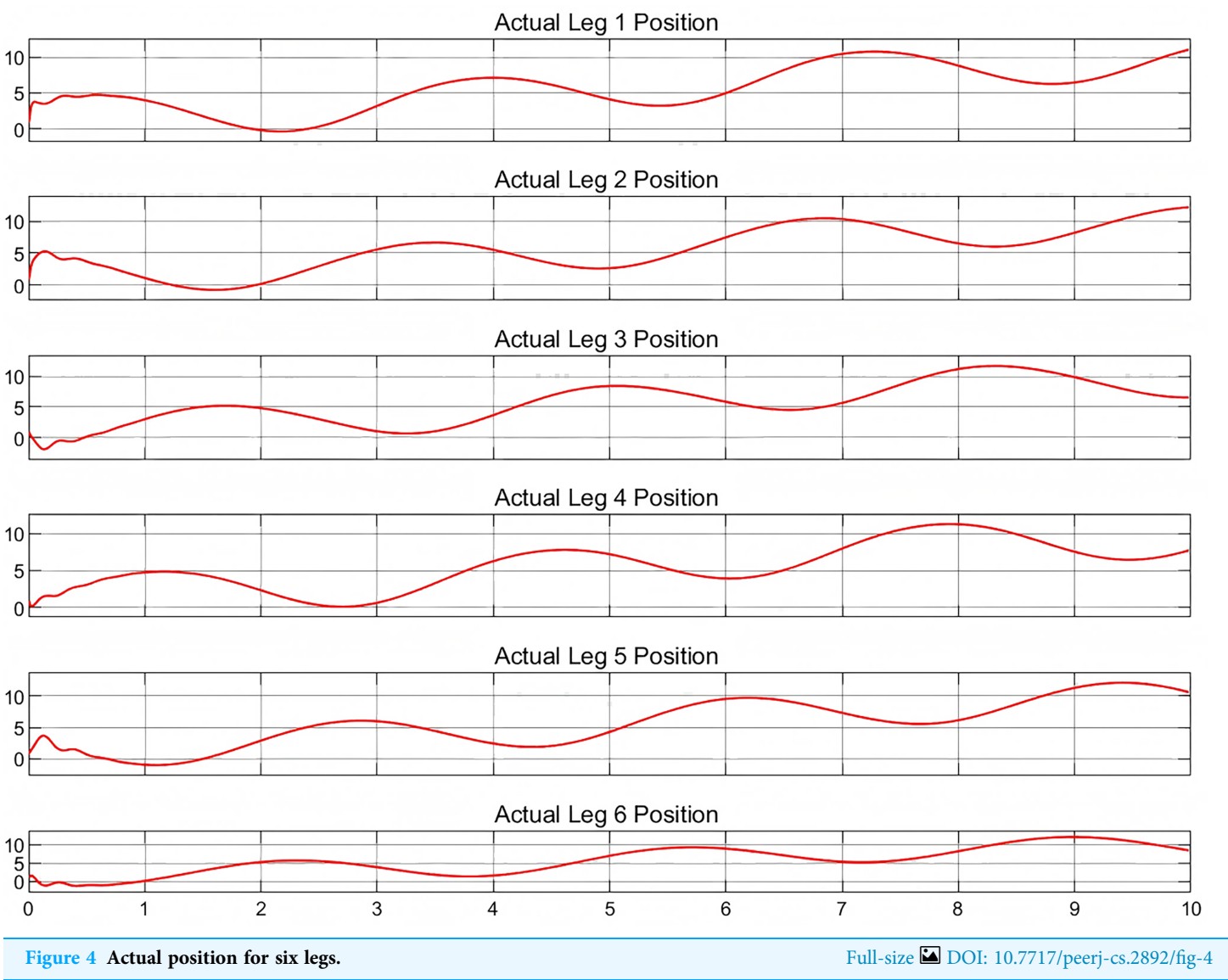

**Figure 4 Actual position for six legs.**

discrepancies between the simulated states and the predefined trajectory, as shown in Figs. 4 and 5, which depict the actual and target positions of the Stewart platform dynamics model, respectively. Furthermore, the calibration of the controller gains revealed the kinematic state error of the model, as shown in Fig. 6. The x-axis represents time in seconds, while the y-axis represents the position error in millimeters. These outcomes were achieved with a proportional gain of 300, an integral gain of 20, and a differential gain of 5. Upon analyzing the images, it is evident that the actual position curves of all the legs closely approximate the desired position curve. This observation suggests that the PID controller, with the current configuration of gains, is effective in ensuring that each leg accurately follows the prescribed path. Additionally, the position error curves reveal that the deviations are minimal, maintaining a variance within ±0.2 cm for the majority of the time. These results corroborate the capability to control with high precision. Furthermore, it is

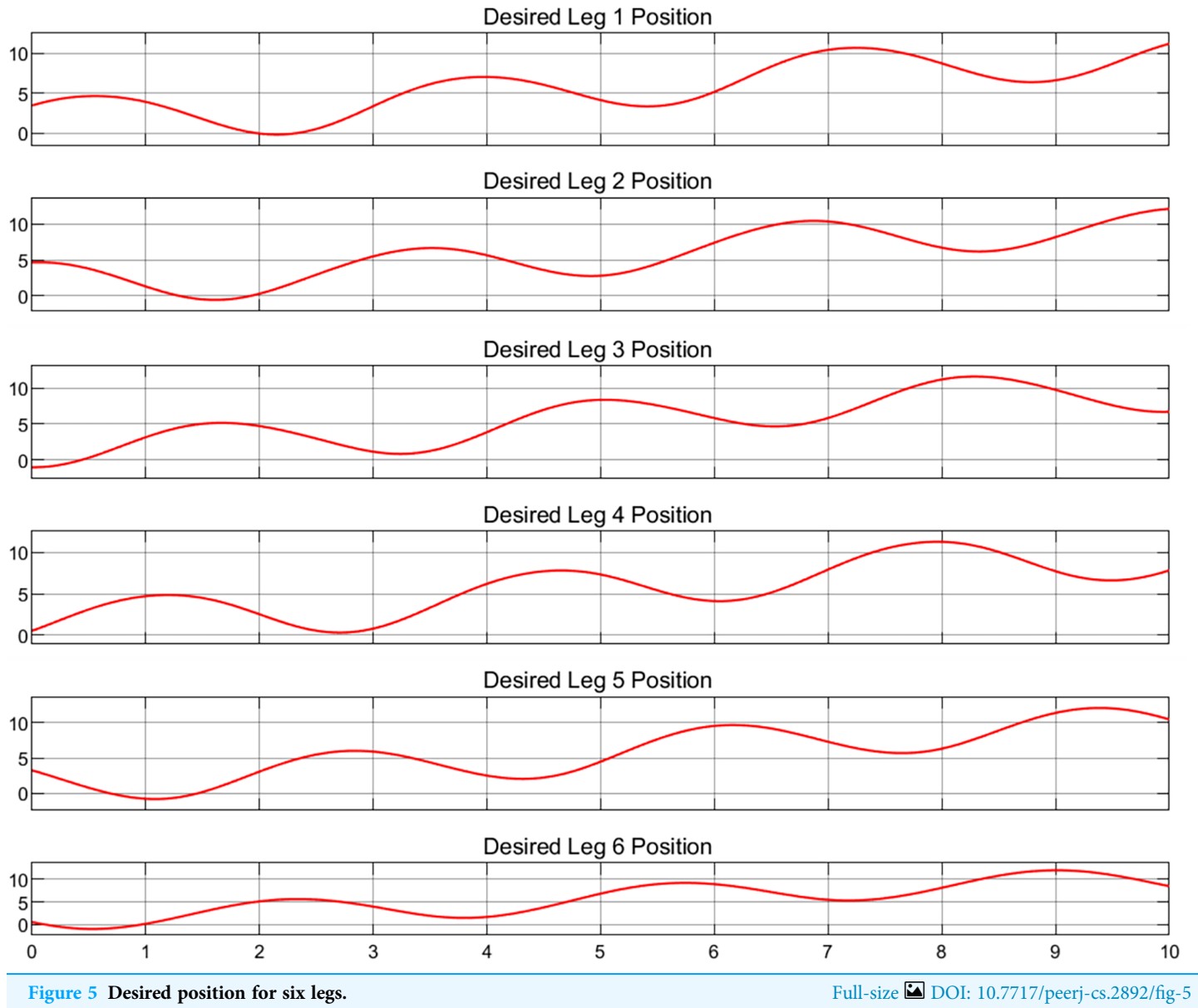

**Figure 5  Desired position for six legs.**     

demonstrated that the simulated dynamics model of the Stewart platform can operate within Simulink using this specific set of gains, ensuring precise and stable performance.

Upon examination of the positional error images, irregular fluctuations are observable at the commencement of each error curve, a phenomenon commonly encountered in such simulations. When the simulation system initiates, the PID controller may experience initial fluctuations due to suboptimal setting of initial conditions or the system requiring time to stabilize. For instance, if the control initial position does not align with the onset of the desired path, additional adjustments by the PID controller are necessary to align the system with the intended trajectory (*Wang et al., 1999*). Additionally, the Stewart platform, as a complex mechanical construct, exhibits dynamic properties such as inertia and

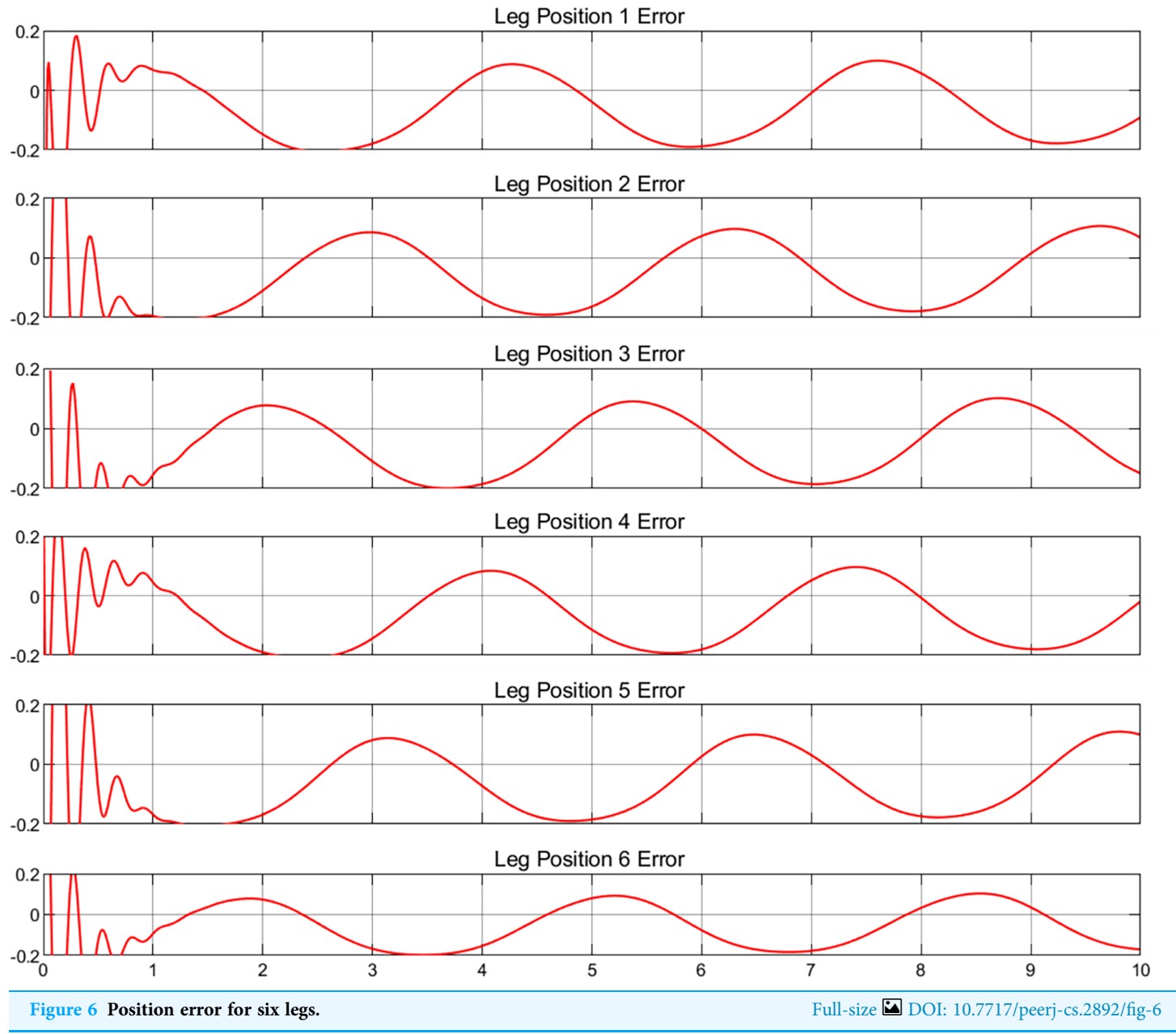

**Figure 6  Position error for six legs.**

damping, which influence the initial response and contribute to error fluctuations at the start of operation. These fluctuations, inherent to the system, cannot be eliminated but can be minimized through careful tuning of the gains (*Mirelez-Delgado, Díaz-Paredes & Gallardo-Carreó, 2020*).

## Digital twin experiment

Following the validation of the multibody model, the initial development phase of the digital twin virtual entity was achieved. However, a complete digital twin model first establishes a reliable bi-directional connection communication to ensure the motion

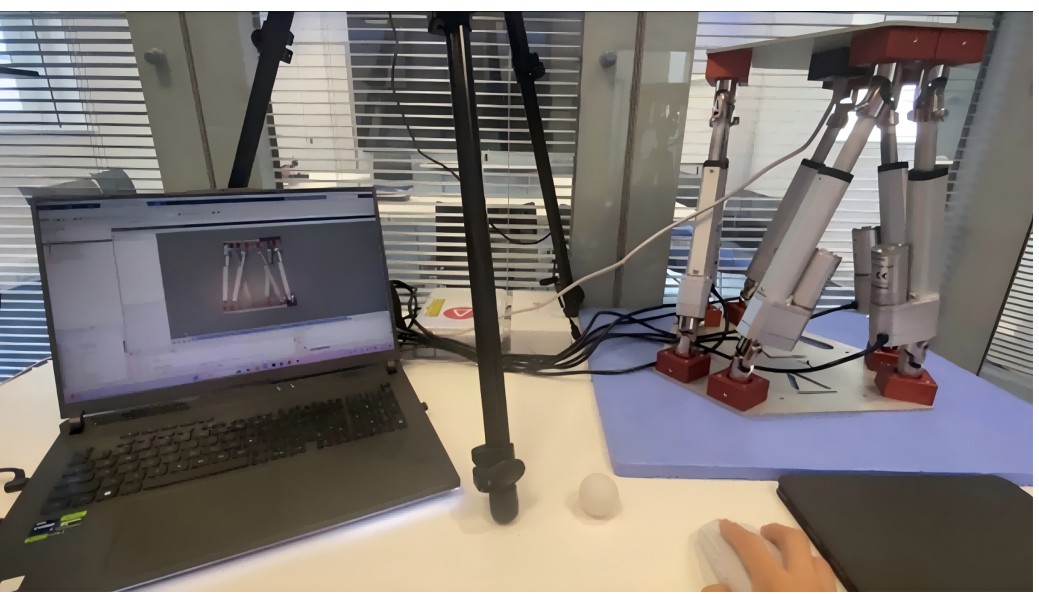

**Figure 7 Digital twin experiment.**

similarity between the virtual end and the physical entity. This section details the results emerging from the bidirectional communication connection between the physical and virtual entities. Additionally, it outlines the challenges encountered during the connection process.

In actual operation, the execution computer carrying the Stewart virtual end needs to be connected to the physical entity *via* a network cable and configured with the same IP address. The operation in the lab was shown in Fig. 7.

In this article, to verify that the digital twin model maintains a satisfactory bidirectional communication link, it is essential to compare the similarity in motion between the virtual end and the physical entity. The physical entity used in the experiments directly outputs the force values representing the extension of the arm. Consequently, data from the physical entity is represented by measured forces. On the virtual side, parameters retrieved from the motion data of the physical entity can be converted into forces using a Jacobian matrix. The Jacobian matrix **J** relates forces and velocities was illustrated below:

$$\vec{F} = J^T\vec{f}, \quad \vec{V} = J\vec{\dot{L}}.$$

Therefore, the processed image in Fig. 8 depicts the forces measured by the virtual end and the physical entity. In this experiment, a given trajectory was simulated at the beginning and commands from the real machine were given after 3 s. The measured forces were processed throughout the experiment.

In the analysis, the red curve represents the force measured at the virtual end, while the blue curve denotes the force measured at the output of the physical entity. Examination of these curves reveals that, for each arm of the Stewart platform, they possess essentially identical shapes. This similarity suggests that the force measurements at the virtual end

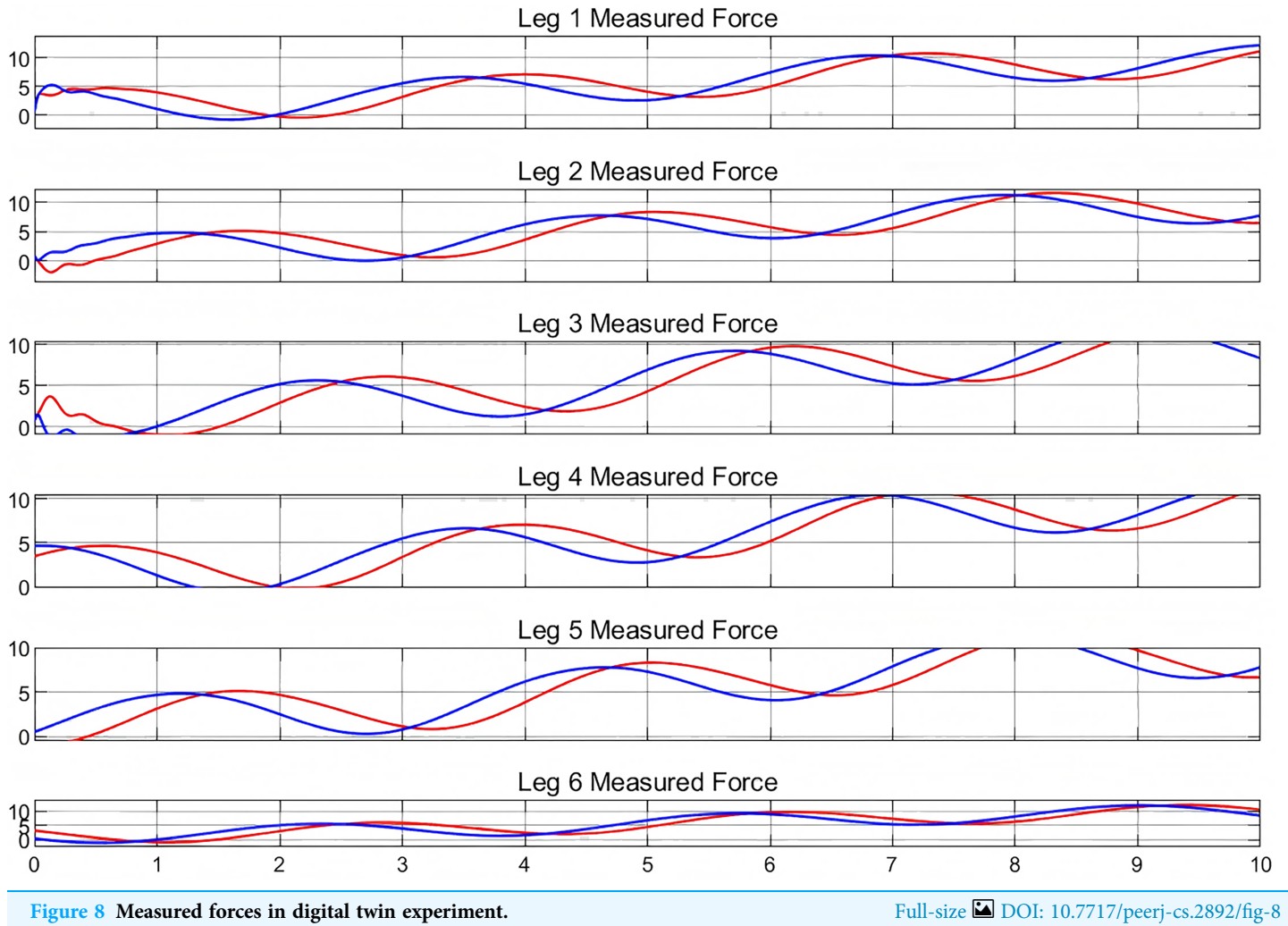

**Figure 8 Measured forces in digital twin experiment.**

correspond closely with those at the physical output, and the extension length of the arm is also consistently comparable. Consequently, the high congruence in motion postures between the virtual end and the physical entity underscores the reliability of the established bidirectional communication connection. This analysis substantiates the efficacy of the implemented digital twin model for the Stewart platform.

Analysis of the graphical data reveals that the curves representing the virtual and physical entities do not coincide, indicating that the virtual entity does not synchronize with the physical entity in real time. The close alignment of these curves confirms the bidirectional communication's accuracy and synchronization. There is a consistent delay between the virtual and physical entities, typically ranging from 3 to 5 s. This delay can be segmented into two distinct components based on prior research. The initial delay originates from the interval between the issuance of the command and its actual execution, approximately a fraction of a second (*Girletti et al., 2020*). The subsequent delay pertains to the measurement and processing within the simulated environment, which is influenced

by the complexity of the simulation (*Tao et al., 2022*). Notably, as the movement trajectory of the Stewart platform increases in complexity, the duration of the delay extends correspondingly. Contemporary research on addressing the challenges associated with delays primarily focuses on the optimization of data communication protocols. Lightweight communication protocols, such as Message Queuing Telemetry Transport (MQTT) and CoAP (Constrained Application Protocol), are specifically designed for IoT devices. These protocols facilitate low-power and low-bandwidth communication between devices and servers, while ensuring low latency and high reliability. Such features render them highly suitable for applications within digital twin scenarios, where efficient and reliable data transfer is critical (*Human, Basson & Kruger, 2021*; *Bhattacharjya et al., 2020*).

In the proposed digital twin system, real-time data exchange between the virtual and physical Stewart platform is facilitated using network communication protocols. Specifically, sensor readings from the physical platform, including actuator positions and force measurements, are continuously transmitted to the virtual model *via* TCP/IP. These signals are processed within MATLAB/Simulink to update the state of the virtual twin in real-time. Moreover, all acquired data is stored in a structured database to enable retrospective analysis and model refinement. The system employs a self-learning mechanism where historical data is used to dynamically adjust the virtual model parameters, ensuring that the digital twin remains up to date and accurately represents the physical system under various operating conditions.

## CONCLUSIONS

This study developed a digital twin model for the Stewart platform by integrating a multibody dynamics simulation with a physical system through bi-directional communication. A comprehensive virtual model was constructed in MATLAB/Simulink, incorporating a trajectory computation module based on the inverse kinematics and Jacobian matrix of the Stewart platform. The key innovation lies in the development of a motion trajectory computation module, which enables real-time transformation of physical input signals into control instructions for the virtual model.

The performance of the simulation model was validated through trajectory tracking and error analysis, confirming its capability to reproduce the dynamic behavior of the Stewart platform with high precision. The developed control strategy, based on tuned PID parameters, demonstrated effective motion tracking with minimal position error. Experimental results further confirmed the consistency between the physical output and virtual response in terms of actuator forces, validating the fidelity of the digital twin model.

### Funding

This work was supported by the Natural Science Foundation of Shanxi Province Grant No. 20210302124257. The funders had no role in study design, data collection and analysis, decision to publish, or preparation of the manuscript.

## Grant Disclosures

The following grant information was disclosed by the authors:
Natural Science Foundation of Shanxi Province: 20210302124257.

## Competing Interests

The authors declare that they have no competing interests.

## Author Contributions

- Xiurui Ding conceived and designed the experiments, performed the experiments, analyzed the data, performed the computation work, prepared figures and/or tables, authored or reviewed drafts of the article, and approved the final draft.
- Jinsheng Xing performed the computation work, authored or reviewed drafts of the article, supervision, and approved the final draft.

## Data Availability

The code is available in the Supplemental File.

## Supplemental Information

Supplemental information for this article can be found online at http://dx.doi.org/10.7717/peerj-cs.2892#supplemental-information.

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
