# Peer review of "Modeling and implementation of a real-time digital twin for the Stewart platform with real-time trajectory computation"

_PeerJ Computer Science, doi:10.7717/peerj-cs.2892_

## Round 0.1 · original submission · Major Revisions

In the opinions of reviewers and mine, this paper should undertake a major revision to address concerns.

Reviewer 1 ·

Basic reporting

The authors provided a thorough assessment of current stat of the art in digital twins. This work describe the development of a robust digital twin of the Stewart platform. Overall, the article is well written and provides ample details in the digital twin development as well as the mathematical underpinning of the Stewart platform. Some comments on your figures:

• Figure 5 is a good figure, but caption does not articulate what is happening. What are the colors meant to express?
• Listing 1, Listing 2 do not seem to add to the discussion; suggest moving to an appendix or removing all together.
• Figures 16,17,18 what are the unites on the x and y axis? You have a set of unites in the title, but not he axes.
• Figure 21; what do the colors mean?
• Figure 22 is a nice figure, but some additional description in the text would be great to describe what all is happening and how you are able to actually create a resilient twin architecture.
• Page 19/24 you have multiple instances of threshoid; should these be threshold? In the Figure 23, it is called threshold.

Experimental design

The one thing that is not clear is how data is passed back and forth between the virtual and physical device; along with this, is data being stored anywhere and used to update any associated models? Some additional description on your resiliency architecture would be appreciated.

Validity of the findings

The validation work is interesting and well described however the description of the actual resiliency portion was lacking. There are no results for your resiliency digital twin. The paper is about a physics based model towards enhanced resilience but you don’t show any results from this. In this regard the conclusions feel weekend by not including enough description or show enough detail about how your approach can be used for resiliency.

·

Basic reporting

The manuscript presents a well-structured and detailed study on developing a resilient digital twin for the Stewart platform. It effectively combines theoretical modeling with practical implementation, demonstrating the potential for enhanced resilience in digital twins. Some minor improvements in clarity and organization could further strengthen the manuscript.

Experimental design

The methodology section is thorough, detailing the experimental setup for validating the digital twin model and the resilient framework. The use of MATLAB/Simulink for modeling and simulation is well-documented.

The procedures for constructing the virtual model and integrating it with the physical platform are well-documented. The manuscript includes detailed descriptions of the experimental setup and the data analysis methods.

Validity of the findings

The results section is clear, presenting the outcomes of the digital twin experiment and the resilient framework. The discussion interprets these results in the context of enhancing resilience.

The discussion connects the results to the research question, explaining how the developed digital twin improves the resilience of the Stewart platform.

Additional comments

The manuscript uses a citation style where a period is placed just before the reference, which is unconventional and may affect the readability of the text. For example, the sentence structure appears as follows: "The concept of a digital twin is increasingly acknowledged as an innovative and promising tool with significant potential in various end-use applications.Singh et al. (2021)"

It would be clearer and more standard to place the reference immediately after the relevant text without an intervening period, like this: "The concept of a digital twin is increasingly acknowledged as an innovative and promising tool with significant potential in various end-use applications (Singh et al., 2021)."

I recommend revising the citation style to improve clarity and consistency throughout the manuscript.

·

Basic reporting

Professional English and Clarity
The manuscript employs professional and clear English throughout, but a few sections have overly complex sentence structures that could be simplified for better readability. For example:

The sentence starting with "This configuration is pivotal in facilitating its capacity..." (Introduction) could be rephrased to enhance clarity.

Some technical Words, such as "Jacobian matrix” and "inverse kinematics," could be briefly explained for a broader audience.


Literature References and Background
The manuscript provides an extensive review of relevant literature, including both recent and foundational studies. However:

A few additional references on current advancements in digital twin applications outside manufacturing (e.g., in healthcare or aerospace) could further enrich the background.

Citations like "Eirinakis et al. (2022)" and "Papacharalampopoulos et al. (2021)" are included but would benefit from a concise explanation of their specific contributions to the field.

Structure, Figures, and Raw Data
The structure aligns with PeerJ standards, with clear sections and logical flow.

Figures are generally relevant, but their quality could be improved. For example, Figure 4 lacks high resolution, making it difficult to distinguish fine details. Figure 14 could use additional annotations to clarify the virtual and physical components of the Stewart platform.

The raw data, including MATLAB and Simulink files, has been provided and is a commendable inclusion. This greatly enhances the reproducibility of the study.

Self-contained and Hypothesis-based
The manuscript is largely self-contained and addresses the stated hypotheses. The description of the "Resilient Digital Twin Framework" is particularly well-articulated, providing readers with a clear understanding of its purpose and implementation.

Experimental design

Original Research within Scope
The research presents an original study on a physics-based model for the Stewart platform within the scope of PeerJ’s computational science criteria. It addresses a critical gap in resilience strategies for digital twin applications, particularly in mechanical systems.

Research Question and Knowledge Gap
The research question is well-defined and meaningful. The authors explicitly identify a knowledge gap in integrating resilience into digital twins for the Stewart platform, which they aim to fill.

Methodological Rigor
The methodology is described in detail, including the use of MATLAB/Simulink for model construction and validation. The availability of raw MATLAB and Simulink data provides transparency and facilitates replication. However:

The validation process for the PID controller lacks clarity regarding parameter tuning and its impact on system stability.

The authors mention using SolidWorks and the Simscape Multibody Link plug-in but do not elaborate on the challenges or limitations of this integration.

Replicability
The availability of MATLAB/Simulink files significantly enhances replicability. Including additional comments within the code, especially for key functions such as the kinematic inverse solution and Jacobian matrix, would further aid reproducibility for future researchers.

Validity of the findings

Data Robustness and Statistical Soundness
The findings are supported by robust data and simulations. For instance, the positional error graphs (Figure 18) effectively demonstrate the accuracy of the PID-controlled system.

The raw data provided aligns with the presented results, confirming the validity of the findings. However, the observed 3-5 second delay in bidirectional communication is inadequately analyzed. This delay could have significant implications for real-time applications, and the authors should provide mitigation strategies or justify its acceptability.

Conclusions and Relevance
The conclusions are well-stated and align with the research questions and results. The emphasis on a resilience framework is timely and relevant, but:

The manuscript’s discussion of future applications (e.g., in other industries) remains speculative and could benefit from specific examples or preliminary results.

Additional comments

The integration of visual monitoring with adaptive thresholds is a notable strength and provides a practical approach to resilience in digital twins. Future work on incorporating advanced AI-based anomaly detection would further enhance this approach.

The manuscript mentions that disruptions will be "created in the experiment by stopping the leg motors," but the specifics of this disruption and its control are not fully detailed.

The potential use of lightweight protocols like MQTT for real-time communication is an excellent suggestion. However, the authors should elaborate on how these protocols could be implemented in the current framework.

Reviewer 4 ·

Basic reporting

The literature references should be enclosed in parentheses for clarity, when they are at the end of the sentences. Otherwise, it may appear confusing, especially when multiple citations appear in the same line, e.g., in line 66, "Tang et al. (2020) Alexios et al."
Figures are sufficient, but texts in Figure 5 need to be enlarged to a larger size, and may need some more texts in its caption.
The caption of Figure 24 needs to highlight its association with the adaptive threshold. The current caption cannot clearly differentiate itself from previous images about equipment.

Experimental design

The experiment is well-performed, but please highlight the difference of the digital twin system from the other system used in the comparative study that leads to the result in Figure 23, and highlight how this system can perform in-time detection and rapid recovery.

Validity of the findings

Figure 16 and 17 are just screenshots of the positional data of legs from Simulink. If they can be compared in one figure for individual legs, it will be clearer to see the differences.
Figure 23 appears confusing, what does the percentage in y-axis refer to? It currently appears the solid blue curve refers to the lost accuracy, and in the initial phase the lost accuracy reaches 90%. Also, what is a criteria for defining the range of adaptive threshold? Will this range ever change to a different experimental condition?

---

## Round 0.2 · Minor Revisions

In the opinions of reviewers and mine, this revised paper should undergo a minor revision.

Reviewer 1 ·

Basic reporting

The authors have adequately addressed concerns I had regarding basic reporting.

Experimental design

The authors have adequately addressed concerns I had regarding basic reporting.

Validity of the findings

The authors have done an excellent job updating their findings.

Additional comments

No additional comments

·

Basic reporting

Professional English Usage:
The manuscript maintains a professional tone and is generally well-written. However, a few minor grammatical errors and awkward phrasings remain. For example, in the introduction, the phrase "the trustworthiness of a digital twin is a critical factor influencing its effectiveness" could be restructured for better clarity.

Literature References and Background:
The authors have done an excellent job providing relevant literature and contextual background for their research. The inclusion of new citations and restructuring of explanations for key terms such as "Jacobian matrix" and "inverse kinematics" improves accessibility for a broader audience.

Figures and Data Presentation:
The authors have addressed my comments regarding figure descriptions. The clarification of color coding in Figure 5 and the addition of axis units in Figures 16-18 enhance clarity. The additional explanation of Figure 22 outlining the three-layer resilience architecture strengthens the manuscript's technical depth.

Self-Containment and Hypothesis Alignment:
The manuscript is self-contained and aligns well with the stated research objectives. However, the conclusions could still benefit from a more explicit discussion on how the study’s findings align with the hypothesis.

Experimental design

Originality and Scope:
The study aligns well with the journal's aims and scope, offering an innovative approach to integrating resilience into digital twins for Stewart platforms.

Research Question and Knowledge Gap:
The research question is well-defined, and the manuscript effectively highlights how it contributes to filling a knowledge gap. The discussion on resilience in digital twins and the integration of adaptive feedback mechanisms strengthens the manuscript’s relevance.

Methodological Rigor and Replicability:
The revised manuscript provides additional details on how data flows between the physical and virtual components. The clarification regarding network communication protocols (TCP/IP) and the use of structured databases for storing and refining model parameters improves transparency and replicability. The explanation of MQTT implementation for real-time communication is a valuable addition.

Validity of the findings

Data and Statistical Soundness:
The authors have responded to concerns regarding validation but should further strengthen their presentation of results demonstrating resilience. The paper primarily focuses on building a digital twin but does not provide extensive quantitative results showing how resilience is improved.

Conclusions and Justification:
While the discussion on resilience mechanisms has been expanded, additional experimental results showcasing the effectiveness of these mechanisms would further solidify the claims. The proposed future integration of neural networks for enhanced resilience is promising, but more evidence from simulations or tests would strengthen the conclusions.

Additional comments

The explanation of how disruptions were introduced (e.g., deactivating Stewart platform actuators) is much clearer now. However, numerical results demonstrating system recovery times or performance improvements would be beneficial.

The discussion section includes future work on integrating AI techniques, such as graph convolutional neural networks, to enhance resilience. While valuable, a brief feasibility assessment or preliminary results would provide additional support for this proposed direction.

---

## Round 0.3 · accepted · Accept

In the opinions of reviewers and mine, this revised paper can be accepted in the current form.

Reviewer 4 ·

Basic reporting

The revision has significantly improved the manuscript. The reduction in the number of figures has enhanced overall readability, and the text is now more concise and focused. The data figures from Simulink are presented in a clearer way. Additionally, the authors have adequately addressed my earlier comments regarding literature citations, providing clearer context and stronger references to prior work.

Experimental design

The research question has clearly defined and addressed a relevant gap in the application of resilient digital twins for Stewart platforms. The integration of modeling, simulation, and real-time physical system validation demonstrates methodological rigor. The experimental design, particularly the use of inverse kinematics, Jacobian computation, and PID tuning, is appropriate and systematically implemented.

Validity of the findings

The findings are generally well supported by the presented data and are aligned with the study’s objectives. The comparison between the physical and virtual systems is clearly illustrated, with PID-tuned trajectory tracking and force measurement validating the digital twin’s accuracy.